# Preparation, Characterization, and Anticancer Effects of Capsaicin-Loaded Nanoliposomes

**DOI:** 10.3390/nu13113995

**Published:** 2021-11-10

**Authors:** Ali Al-Samydai, Walhan Alshaer, Emad A. S. Al-Dujaili, Hanan Azzam, Talal Aburjai

**Affiliations:** 1Diagnostic Research Centre, Department Pharmacological, Faculty of Pharmacy, Al-Ahliyya Amman University, Amman 19328, Jordan; a.alsamydai@ammanu.edu.jo; 2Cell Therapy Center, The University of Jordan, Amman 11942, Jordan; walhan.alshaer@ju.edu.jo; 3Centre for Cardiovascular Science, Queen’s Medical Research Institute, University of Edinburgh, 47 Little France Crescent, Edinburgh EH16 4TJ, UK; 4Hamdi Mango Center for Scientific Research (HMCSR), University of Jordan, Amman 11942, Jordan; hanan-azzam1@hotmail.com; 5Department of Pharmaceutical Sciences, Faculty of Pharmacy, University of Jordan, Amman 11942, Jordan

**Keywords:** anticancer, *Capsicum annuum*, capsaicin, nanoliposomes, polymer-based nanocarriers, release kinetics

## Abstract

Background: Medicinal plants have proven their value as a source of molecules with therapeutic potential, and recent studies have shown that capsaicin has profound anticancer effects in several types of human cancers. However, its clinical use is handicapped due to its poor pharmacokinetics. This study aims to enhance capsaicin’s pharmacokinetic properties by loading the molecule into nanoliposomes model and testing its anticancer activity. Methods: Nanoliposomes were prepared using the thin-film method, and characteristics were examined followed by qualitative and quantitative analyses of encapsulation efficiency and drug loading using HPLC at different lipid/capsaicin ratios. Cell viability assay (MTT) was used to determine IC_50_. Results: Capsaicin-loaded nanoliposomes showed optimum characteristics of morphology, particle size, zeta potential, and stability. In vitro anticancer activity of capsaicin and capsaicin-loaded nanoliposomes were compared against MCF7, MDA-MB-231, K562, PANC1, and A375 cell lines. Capsaicin-loaded nanoliposomes showed significant improvement in anticancer activity against cancers cell lines studied (*p* < 0.001), with increased selectivity against cancer cells compared to capsaicin. Conclusion: The encapsulated capsaicin nanoliposomes produced an improvement in pharmacokinetics properties, enhancing the anticancer activity and selectivity compared with capsaicin. This model seems to offer a potential for developing capsaicin formulations for the prevention and treatment of cancer.

## 1. Introduction

Cancer is the second leading cause of mortality worldwide and cancer resistance has become a global crisis. WHO has stated that >18 million people have been diagnosed with cancer, and >9.6 million people have died due to cancer in 2017 worldwide! The incidence of cancer has been growing alarmingly; in 2014, >1.6 million people had cancer, with around 1.2 million reported deaths due to cancer between 2014 and 2015 in the USA alone [1]. Cancer represents a real crisis affecting the health of all humans. Unfortunately, it is a diverse disease at the tissue level, and this diversity presented a challenge in the specificity of diagnosis and efficacy of treatment against different types of cancer [1,2]. Research in cancer diagnosis and therapy is ongoing continually, and our knowledge of cancer characteristics and behavior are updated daily. Cancer is characterized by uncontrolled growth, cell division, spread of heterogeneous group of cells and alteration in the dynamic genome, which led to the production of carcinogenic characteristic in normal cells and termed as cancer, malignant, or tumor cells [3,4].

Cancer cells have the ability to develop alternative pathways to survive and propagate in spite of therapies, by producing ways of resistance to treatment [5]; as a result, chemotherapy resistance (multiple drug resistance) has become the biggest challenge in cancer therapy, which posed a major impediment to patient survival and is the primary cause of patient death in most advanced stages of cancers [6]. Several factors can lead to tumor therapeutic failure including genetics, epigenetics, and microenvironment factors [7], and these have shown an increasing efflux of drug permeability and enzymatic deactivation, making cancer cells able to stay alive regardless of the presented chemotherapy [8]. Most of the anticancer drugs have also shown poor selectivity for cancerous cells over healthy cells due to low solubility in aqueous fluids and poor cellular uptake, which leads to low therapeutic efficiency and/or unwanted side effects [9,10,11]. In addition, there are no magic treatments for cancer that are safe, effective, and low cost which are convenient for long-term use with minimum toxicity. For these reasons, there is an indigent need to develop a novel therapeutic choice of increasing the efficacy and reducing the side effects by increasing selectivity towards cancer cells over normal cells [12].

Even though the pharmaceutical industry in the last century has shifted its focus toward synthetic compound libraries and high throughput screening for the discovery of new drug leads [13], numerous medicinal plant compounds are found to be an important source for many clinically used anticancer agents such as vincristine, vinblastine, topotecan, irinotecan, etoposide, and paclitaxel (Taxol^®^) [14]. Capsaicin (CAP) is a naturally occurring alkaloid derived from chilies (*Capsicum annuum* (*C. annuum*), family, Solanaceae) that is responsible for its hot pungent taste. It is an odorless fat-soluble compound [15], and lately, researchers have found that CAP targets multiple signaling oncogenes pathways, and tumor suppressor genes in a variety of types of cancer models. CAP has been shown to induce apoptosis in several types of cancer cell lines including bladder, breast, colonic, esophageal, leukemia, lung, liver, pancreatic, prostatic, skin, while leaving normal cells unharmed [16]. In addition, the latest review by Bley et al. noted that CAP appears to induce apoptosis in over 40 distinct cancer cell lines [17].

The clinical use of CAP was handicapped by its quick first-pass metabolism and a short half-life of less than 8 min following intravenous administration, and such poor oral bioavailability is mainly due to its poor aqueous solubility [18,19]. Furthermore, CAP is an extremely irritating compound causing pain and burning of skin, mucosa resulting in gastrointestinal side effect even at low concentrations [20]. Liposomes are an important example of targeted drug delivery which improve the therapeutic index for many drugs by stabilizing the compounds, increasing the drug concentration and the residence time in target cells by overcoming obstacles to cellular and tissue uptake [21]. Liposomes, as a drug delivery system for cancer cells, present numerous advantages including the ability for self-assembly, capacity to carry large drug payloads, biocompatibility, and possess a large range of biophysical and chemical characteristics that could be modified to control and improve the drug biological properties [21,22,23]. Nanoliposomes were successfully translated into clinical applications as drug delivery systems (DDs) firstly in 1965. Recently, several research projects were carried out using these nano DDs in various fields of therapies. The important role of liposomes as DDs in the healthcare secta is recognized by FDA-approved liposomal drug formulations, e.g., Doxil^®^, Epaxal, DaunoXome Ambisome^®^, DepoDur™, Depocyt, etc. [24,25].

The current study aimed to load the CAP into nanocarrier (PEGylated liposomes), followed by characterization of CAP-loaded nanoliposomes preparations for size, zeta potential (charge), encapsulation efficiency (%EE), drug loading (%DL) and stability. The anticancer activity of CAP and CAP-loaded nanoliposomes were investigated and compared in vitro against different types of cancer cell lines.

## 2. Materials and Methods

### 2.1. Chemical Reagents

Chloroform (Acros organic, Geel, Belgium), ethanol, methanol (Fisher Scientific, Waltham, MA, USA), and acetonitrile (Honeywell, Brandenburg, Germany); CAP standard (Santa Cruz Biotechnology, Dallas, TX, USA). Deuterated Dimethyl sulfoxide (d6-DMSO) (99.9% atom) was purchased from Sigma-Aldrich, St. Louis, MO, USA. 1,2-Dipalmitoyl-sn-glycero-3-phosphocholine (DPPC), Cholesterol (CHOL), and DSPE-PEG 2000 [1,2-distearoyl-sn-glycero-3-phosphoethanolamine-N-[amino(polyethylene glycol)-2000] were obtained from Avanti Polar Lipids, Inc. (Alabaster, AL, USA). Phosphate buffered saline (PBS) was obtained from Lonza Group Ltd., Basel, Switzerland. De-ionized water used to prepare the aqueous solutions throughout the experiments was obtained from RiOsTM type 1 simplicity 185, Millipore Waters, Temecula, CA, USA. 

MCF-7 (ATCC number: HTB-22) and MDA-MB-231 (ATCC number: HTB-26) cells, A375 (ATCC number: CRL-3222), PANC1 (ATCC number: CRL-1469), A569 (ATCC number: CCL-185) and Fibroblast (ATCC number: PCS-201-012) were obtained from the Hamdi Mango Center for Scientific Research (The University of Jordan). All cell lines were cultured in RPMI 1640 growth medium (RPMI) (Capricorn Scientific GmbH, Ebsdorfergrund, Germany) and DMEM, EMEM (Euroclone SpA, Via Figino, Italy); supplemented with 10% (*v*/*v*) fetal bovine serum (FBS), 1% (*v*/*v*), 200 mM L-glutamine, and antibiotics; Penicillin-Streptomycin (100 IU/mL-100 µg/mL), respectively. Accutase purchased from Capricorn Scientific GmbH, Ebsdorfergrund, Germany. Cell culture plates from TPP, Switzerland and Agarose gel from Promega, Madison, Wisconsin, USA. Acetone, acetonitrile, and 1-propanol were obtained from DMG Carbon group, London, England. Formaldehyde from Leica, Buffalo Grove, Illinois, USA. All other chemicals and solvents were of analytical grade and were used without further treatment.

### 2.2. Preparation and Characterization of Nanoliposomes

Nanoliposomes were prepared by thin-film hydration method as previously described [26,27]. Three different formulae were prepared, as shown in Table 1 and Table 2, to study the effects of various lipid ratios (DPPC:Cholesterol:DSPE/PEG2000), F1; 85:10:5, F2; 75:20:5, F3; 65:30:5 with fixing CAP concentration at 1.67 mg/mL was chosen according to the literature [26]. The optimum formula with ratio of 75:20:5 (DPPC:Cholesterol:DSPE/PEG2000) tested for its maximum encapsulation efficiency with the concentration of lipid was 0.51 mmole. Four more concentrations were also tested by changing the amounts of CAP (F4: 0.835, F5: 3.34, F6: 6.68, and F7: 0.00 mg/mL), to determine the maximum encapsulation efficacy, but they were found not suitable.

#### Characterization of Nanoliposomes

Size and Zeta potential (Charge)

Average size, charge, and polydispersity index (PDI) for free liposomes and loaded liposomes were measured by dynamic light scattering (DLS) on a Zetasizer (Malvern Instruments Ltd., Malvern, UK). Samples were diluted 1:20 with distilled water [26,27].

b.Encapsulation efficiency (%EE) and drug loading (%DL)

The percentage of CAP encapsulated was determined by degradation of liposomes and was obtained by the addition of 800 µL HPLC grade acetonitrile to 200 µL of the liposomes, followed by 10 min sonication at 35 °C, then 10 min centrifugation at 12,000 rpm. The filtrate was filtered using a 0.45 µm syringe filter. The concentration of encapsulated CAP was measured by an HPLC system (UV-VIS-PDA Detectorat 220 nm, aKromasil^®^ C18 column); with diameters 150 mm × 4.6 mm, 5 µm, a flow rate of 0.5 mL/min and column temperature 40 °C using a 20 µL injection volume connected to computer software Chromeleon^®^. The mobile phase used is 80:20 Methanol: Acetonitrile respectively [27].

The %EE and %DL were calculated as:$$\left(\%\mathrm{EE}\right)=\frac{\left[\mathrm{Entrapped}\;\mathrm{CAPs}\right]}{\left[\mathrm{Total}\;\mathrm{CAPs}\right]}\times 100$$
$$\left(\%\mathrm{DL}\right)=\frac{\left[\mathrm{Weight}\;\mathrm{of}\;\mathrm{loaded}\;\mathrm{drug}\right]}{\left[\mathrm{Weight}\;\mathrm{of}\;\mathrm{lipids}\right]}\times 100$$

c.Drug release

The release experiment was done in triplicate. In three Amicon^®^ filters with a cut-off of 100 kDa (Millipore (U.K.) Limited, Livingston, UK), 750 µL liposomes were taken in portions and centrifuged at 8000 rpm for 10 min to remove free drug and concentrated to 400 µL, then transferred to Eppendorf tubes, and the volume was completed up to 2 mL with PBS (pH 7.4); the three tubes were incubated at 37 °C with shaking. Samples were taken at different time points (0, 15, 30 min, 1, 2, 3, 4, 24, and 48 h), and at each time point, 150 µL were drawn and centrifuged at 8000 rpm for 10 min to remove free CAP. After 48 h, 600 µL of acetonitrile were added to all the samples to destroy the liposomes as mentioned before, then these samples were analyzed using HPLC to determine the unreleased CAP. Then, the percent release was calculated according to the following equation [25,27]:$$\%\;Release=\frac{concentration\;at\;t0-concentration\;at\;t\;}{concentration\;at\;t0}\times 100$$

d.Transmission electron microscopy (TEM)

Structure and morphology of empty and CAP-loaded liposomes were analyzed by TEM. TEM analysis of liposomes samples was performed using the negative staining method. First, 200-mesh formvar copper grids (SPI supplies, West Chester, PA, USA) were coated with carbon under a low-vacuum Leica EM ACE200 glow discharge coating system (Leica, Westbahnstraße, Austria). Then, the carbon-coated grids were further coated with 1.5% Vinylec K solution in chloroform. A drop of deionized water-diluted liposomes suspension was placed on the 200-mesh copper grid followed by air-drying. The loaded grids were then stained with 3% *v*/*v* of aqueous solution of uranyl acetate for 20 min at room temperature. After incubation, grids were washed with distilled water and dried at room temperature before imaging with Versa 3D (FEI, Hoofstraat, The Netherlands) TEM operating system at an acceleration voltage of 30 KV [27].

### 2.3. Cell Viability Assay (MTT)

A375, MCF7, PANC1, MDA-MB-231, Fibro cell lines (5 × 10^3^ cells per well) and K562 (30 × 10^3^ cells per well) were seeded in 96-well plates (TPP, Zollstraße, Switzerland). After 24 h, cells were treated with different concentrations of CAP, and liposome loaded with CAP, and without any treatment as negative control; then incubated at 37 °C for 72 h. Then treatments were replaced with 15 μL of 3-(4,5-Dimethyl-2-thiazolyl)-2,5-diphenyltetrazolium Bromide (MTT) solution (Bioworld, Visalia, CA, USA) and 100 μL of medium RPMI used for A375, K562, MCF7, and Fibro, while DMEM media used for Panc1 and MDA-MB-231. After incubation for 3 h, the medium was removed, and the cells were mixed with 50 μL of dimethyl sulphoxide (DMSO). The absorbance was measured at a wavelength of 570 nm using a Glomax microplate reader (Promega, Madison, WI, USA).

### 2.4. Statistical Analysis

The results were presented as the mean ± standard deviation of at least three independent experiments. Statistical significance was determined by using different statistical tests (normality test, Pearson correlation, paired *t*-test, and one-way ANOVA). A value of *p* < 0.05 was considered to assign a statistically significant difference. SPSS software, Version 21, GraphPad Prism 6 (GraphPad Software Inc., La Jolla, CA, USA), and Microsoft Office Excel 2010 (Microsoft, Washington, DC, USA) were used.

## 3. Results

### 3.1. Characterization of Nanoliposomes

#### 3.1.1. Effect of Lipid Ratio on Particle Size, PDI, Zeta Potential and EE

CAP-loaded nanoliposomes were formulated by using different ratios of Lipids (DPPC: Cholesterol: DSPE/PEG2000), F1-85:10:5, F2-75:20:5, F3-65:30:5, respectively. The particle size, PDI, Zeta potential, and %EE of the different formulation is shown in Table 1. The specificity test was to guarantee that CAP was reliably determined, eliminating the possibility of false-positive results due to encapsulation of CAP in liposomes and constituents of the matrix used in the test or due to the decomposition-derived elements, as shown in Figure 1, chromatograms of CAP standard, and control blank sample, of PBS, the retention time was 3.22 and 2.76 min of CAP and PBS, respectively.

The mean ± SD particle size, PDI, zeta potential and EE of all formulations are given in Table 1 and are within the range of nanoliposomes optimum formulation [28,29], where the minimum recorded data of particle size, PDI, zeta potential and %EE was 94.57, 0.05, −17.50 and 13.54%, respectively, and maximum recorded data of particle size, PDI, zeta potential and EE were 149.00, 0.19, −11.50, and 32.10%, respectively.

The results showed that the average particle size of three formulae CAP-loaded nanoliposomes were different with *p* = 0.006, “Multiple comparisons were performed using the LSD procedure at α = 0.05”. Whereas mean size of F1 (99.48 ± 3.90 nm) was significantly smallest than the mean particle size of F2 (118.54 ± 14.18 nm) and F3 (123.68 ± 22.72 nm), with *p* = 0.009 and 0.003, respectively, the mean particle size of F2 (118.54 ± 14.18) showed no significant difference in compassion with F3 (123.68 ± 22.72 nm), as shown in Table 1. The relationship between particle size/ratio of CHOL for the average PDI, ZETA and %EE were found to be non-significant.

Our results showed that F2 had the highest mean of encapsulation (22.38%), with maximum encapsulation up to (32.10%). Based on collected data particle size, PDI, Zeta potential and encapsulation efficacy, we chose F2 as a representative and an optimized formula for further studies.

The TEM study reveals that CAP-loaded nanoliposomes F2 has been done and data represented a spherically shape and uniformly size of CAP-loaded nanoliposomes with an average particle size of around 60.65 ± 19.5 nm (n = 15). Figure 2B,C shows clearly that CAP has been encapsulated in the bilayer of liposomes.

#### 3.1.2. Effect of Capsaicin Amount on Particle Size, PDI, Zeta Potential, %EE and %DL

The mean ± SD particle size, PDI, and zeta potential of all formulations are given in Table 2. All data are within the range of nanoliposomes optimum formulation [28,29].

Results showed that the average size, PDI and Zeta potential of nanoliposomes for the five formulas (F2, F4, F5, F6, and F7) were not significant, with *p* values of 0.328, 0.522 and 0.488, respectively, which means that the CAP amount had no impact on particle size, PDI and Zeta potential. Whereas the CAP amount had an impact on %EE and %DL, as shown in Figure 3. One-way ANOVA of results found that there was a significant difference across different formulae (F2, F4, F5, and F6) with *p* < 0.001. A multiple comparison was performed using LSD procedure at α = 0.05, where %EE data represented that F2 (mean = 22.37 ± 6.02) was significantly higher %EE than F5 (mean = 10.65 ± 0.844) and F6 (mean = 4.87 ± 1.165) with *p* < 0.001. On the other hand, F2 showed no significant difference in comparison with F4 (M = 20.16 ± 1.23) with *p* = 0.278. Additionally, F4 (mean = 20.16 ± 1.23) has shown significantly higher %EE in comparison with F5 (mean = 10.65 ± 0.844) and F6 (mean = 4.87 ± 1.165) with *p* < 0.001, toward F5 and F6. Also, the mean %EE of F5 (mean = 10.65 ± 0.844, N = 6) was significantly higher than the mean %EE of F6 (mean = 4.87 ± 1.165, N = 6) with *p* = 0.019, then Pearson’s correlation coefficient had been done to investigate the relationship between two variables (CAP amounts and %EE), as shown Figure 3.

The results showed that we had a strong negative correlation as CAP amount increases in value, the %EE decreases in value with *p* value (2-tailed) at <0.001 and Pearson correlation of −0.828, which means that we have a reverse relationship between CAP amount and %EE. Furthermore, when the impact of CAP amount on %DL was studied, data showed a significant difference across different formulas (F2, F4, F5, and F6) with *p*-value < 0.001. A multiple comparison was performed using LSD procedure at α = 0.05, where %DL data represented that F2 (mean = 0.75 ± 0.20), F5 (mean = 0.71 ± 0.06) and F6 (mean = 0.65 ± 0.15) were significantly higher %DL than F4 (mean = 0.34 ± 0.20) with *p* value < 0.001, <0.001, and <0.001, respectively. Whereas, F2 showed no significant difference in comparison with F5 and F6 with *p* value = 0.637 and 0.197, respectively. Besides, F5 and F6 showed no significant difference with *p* = 0.470. From the data above, we found that F2 showed the highest mean of %DL, and as we increase CAP amount, there was a slight decrease in %DL, as shown in Figure 3. Maximum encapsulation of CAP inside liposomes in F2 was reached, with highest %DL and any increase in CAP amount could cause a reduction in %EE and %DL.

#### 3.1.3. Stability of Liposomal Formulations

The stability of particle size, PDI and zeta potential of CAP-loaded liposomal formula F2 was evaluated at 4 °C storage, at intervals (0, 14 days, 30 days, 60 days, and 120 days). The mean ± SD particle size, PDI, and zeta potential of formula F2 during stability study results were within the range of nanoliposomes optimum formulation [28,29], where the minimum recorded data of particle size, PDI, and zeta potential were 98.01, 0.09, and−17.5, respectively, and maximum recorded data of particle size, PDI, and zeta potential were 179.30, 0.37, and −6.68, respectively. The physical stability of F2 Formula was evaluated at 4 °C, and results showed that CAP-loaded liposomes were stable up to 4 months at refrigerator temperatures as the particle size, PDI and Zeta potential of the liposomes did not change during this period.

When the in vitro release of CAP was studied in physiological buffer (PBS, pH 7.4) at 37 °C for 48 h at different intervals, from CAP-loaded nanoliposomes, it was found that about 24% release of drug was obtained from the CAP-loaded nanoliposomes (F2) after 48 h (Figure 4). The results indicated that CAP showed a biphasic release from CAP-loaded nanoliposomes, with fast release at the first hour (12%), followed by sustained release up to 48 h (24.0%). The initial fast release in the biphasic behavior can be explained by the release of CAP trapped near liposomes membranes, as shown in Figure 4.

### 3.2. In Vitro Anticancer Activity (Cell Viability Assay)

To establish a critical role of nanoliposomes on anticancer activity of CAP, the cellular response has been investigated against five different types of cancer cell lines (MCF7, MDA-MB-231, A375, K562 and PANC-1). Different ratios of CAP and CAP-loaded nanoliposomes have been used to determine the concentration (μM) that can inhibit 50% of cell growth (IC_50_) of MCF7, MDA, A375, K562, and PANC-1 and results are shown in Figure 5. A paired *t*-test of the mean value of IC_50_ between treatments (CAP IC_50_ = 515.55 ± 207.8 μmole) and CAP-loaded nanoliposomes (IC_50_ = 21.52 ± 10.90 μmole) with *p* = 0.005, indicated that there is a significant difference between groups, and an improvement in anticancer activity against human cancer cell lines (MCF7, MDA-MB-231, K562PANC-1 and A375) compared to free CAP. The toxicity of CAP and CAP-loaded nanoliposomes was investigated against human normal cell fibroblasts (negative control), as shown in Figure 6 and Figure 7 and Table 3, respectively. One-way ANOVA was used for comparing the results of mean IC_50_ of CAP and CAP-loaded nanoliposomes against human cancer cell lines (MCF7, MDA-MB-231, A375, K562 and PANC-1). Table 3 shows that there was a significant difference in IC_50_ of CAP and CAP-loaded nanoliposomes between normal and cancer cells, with *p* < 0.001. Multiple comparisons were performed using LSD procedure at α = 0.05, where data of IC_50_ values for CAP and CAP-loaded -nanoliposomes of normal human fibroblast cells were significantly higher than IC_50_ values of CAP and CAPloaded nanoliposomes against cancer cells with *p* < 0.001.

From these results, we concluded that CAP and CAP-loaded nanoliposomes showed good selectivity towards cancer cell (MCF7, MDA-MB-231, A375, K562 and PANC-1) in comparison with normal cell Fibro, with (1.3, 1.2, 4.1, 1.8, and 1.1) and (4.9, 1.7, 5.0, 3.7, and 2.9) fold respectively.

A paired *t*-test had been done to test the impact of loaded CAP in nanoliposomes in improving the selectivity of CAP towards cancer cells and the following hypotheses had been developed to test this claim:

H0: Md = 0 (the mean fold of selectivity against cancer cells is zero; the loaded CAP in nanoliposomes has no impact in improving selectivity).

Ha: Md < 0 (the mean fold of selectivity against cancer cells is negative; the loaded CAP in nanoliposomes has a significant impact in improving selectivity).

The mean fold CAP IC_50_ selectivity (mean = −1.79 ± 1.18, N = 5) was significantly lower than zero, t (4) = −3.23, one-tailed *p* = 0.032 < 0.05, providing evidence that the loaded CAP into nanoliposomes is effective in producing higher selectivity, as shown in Table 3.

## 4. Discussion

Plants have been used by humans as medicines by ancient civilizations. Since then, traditional medicine has been developed and incorporated in therapy as important sources of drug discovery [30,31]. Recent research has demonstrated that CAP has anti-inflammatory, anticancer, antidiabetic, anticoagulant and hypolipidemic activity [16].

Encapsulation and characterization studies of CAP-loaded nanoliposomes were dedicated to formulating an optimum nanoliposome carrier for CAP to evaluate the anticancer activity against a group of cancer cell lines (MCF7, MDA-MB-231, A375, K562, and PANC-1). Liposomes with the size of 118.54 ± 14.41 nm were prepared and composed of DPPC as a neutral phospholipid, CHOL added to the bilayers mixture to act as a fluidity buffer, intercalator with phospholipids molecules, alters the freedom of formation of carbon molecule and for lowering membrane permeability, and imparting better stability of liposomes, and DSPE-PEG to confer “stealth” properties to the liposomes [32]. To our knowledge, there is no study yet describing the anticancer activity of CAP-loaded nanoliposomes against several cancer cell types in vitro. In this study, the improvement of anticancer activity after loading of CAP in the nanoliposomes model showed consistent results with previous studies where different encapsulated compounds were used [33,34,35,36]. Liposome’s size, charge, polydispersity index, %EE and TEM images indicate the presence of a homogeneous population before and after CAP loading, with optimum nanoliposomes formulations [28,29]. It was found that positive relationship between particle size and the ratio of CHOL in our formula; as we increase the percentage of CHOL, particle size was increased, while the average PDI and ZETA were not affected. Formula F2 was found to be the best produced formula with the highest mean of %EE and %DL (22.37 and 0.75, respectively). The TEM study reveals that CAP-loaded nanoliposomes F2 were spherical in shape and uniform in size with an average particle size of around 60.65 nm (n = 15), and that CAP was clearly encapsulated in the bilayer of liposomes. The physical stability of F2 Formula was evaluated, and results showed that CAP-loaded liposomes were stable up to 4 months at 4 °C (particle size, PDI and Zeta potential of the liposomes did not change during this period). CAP release from encapsulated liposome in vitro showed an initial fast drug loss of 8.40% after 15 min, followed by slower rates of drug loss (24.06% after 48 h) and this is compatible with other studies [33], where CAP is mainly associated within the bilayer lipid structure of the liposomes. In addition, the in vitro release study of CAP showed no burst effect allowing the drug transport out of the liposomes to be driven mainly by a diffusion-controlled mechanism. This suggests that lipid bilayer was stabilized by cholesterol and thus takes time for CAP to be released from the loaded liposomes [37]. There are several benefits of choosing nanosystems over the micro-systems such as longer circulation time in the blood stream without being recognized by macrophages, simplicity of penetration into tissues through capillaries and biological membranes, the capability to be taken up by cells effortlessly, offering high therapeutic activity at the target site, sustaining the effect at the desired area over days or even weeks, improving controlled release and precision targeting of the entrapped compounds to a greater extent. Nanoliposomes can also amplify the performance of bioactive agents by improving their solubility and bioavailability [38].

The toxicity of CAP-loaded nanoliposomes was investigated against human fibroblasts as negative control; IC_50_ value of CAP-loaded nanoliposomes against normal fibroblast cells showed higher value than human cancer cell lines (MCF7, MDA, A375, K562 and PANC-1) with 4.9, 1.7-, 5.0-, 3.7-, and 2.9-fold increase, respectively, and increased the selectivity of CAP nanoliposomes against cancer cells in comparison with free CAP. This was thought to be due to increased bioavailability of poorly water-soluble CAP, increased cytotoxicity of CAP, and the fact that nano-liposome exhibited a slower release behavior, which prevented the burst release of the drug and reduced its hemolytic toxicity, and relevant antitumor activities [35]. Therefore, CAP-loaded nanoliposomes showed improvement in capsaicin’s anticancer activity and selectivity. These results were compatible with previous studies as liposomal formulations increase drug accumulation in tumor cells as well as intracellular drug retention, in vitro and in vivo [24].

The transient receptor potential cation channel subfamily V member 1 (TRPV1) which is also known as the capsaicin receptor, is an overexpression protein in many cancers types whose function might be critical in various neuronal physiological conditions. The epidermal growth factor receptor (EGFR) is a receptor tyrosine kinase that is over expressed in many human epithelial cancers and is a potential target for anticancer drugs [39,40]. TRPV1 interacts with EGFR, leading to EGFR degradation. Capsaicin inhibited the EGFR-induced invasion and migration of human tumor cells [39]. These studies showed that capsaicin’s role as a potent antimetastatic agent, which can markedly inhibit the metastatic and invasive capacity of tumor cells, with an increase in the selectivity of capsaicin and capsaicin-loaded nanoliposomes against cancer cells above normal cells [41].

Many phytochemicals showed challenging anticancer activities for the prevention and/or treatment, due to their low aqueous solubility, poor stability, unfavorable bioavailability, and low target specificity [36]. There is now great demand to develop novel DDs strategies from these natural products, and liposomes nanoparticles showed a good model for phytochemicals delivery, for its biocompatibility and biodegradability. These liposomal nanoparticles could enhance the solubility and stability of phytochemicals, increase their absorption and bioavailability, protect against premature enzymatic degradation or metabolism, prolong their half-life, and improve their target specificity to cancer cells or tumors via passive or targeted delivery, and lower toxicity or side-effects to normal cells or tissues. With sustained-release formulations to prevent prematurely interacting with the biological environment and enhancing anti-cancer activities, nanoliposomes offer a great potential for developing phytochemical-loaded nanoparticles [42]. These can be given by many routes of administration, where parenteral administration is the predominant one for clinically approved products in addition to ocular, and transdermal routes to achieve the desired clinical success [43].

## 5. Conclusions

Our results showed that novel CAP-loaded nanoliposomes produce good stability, selectivity, and safety and offer higher activity than CAP alone against cancer cells. These results were compatible with other anticancer agents using the encapsulated nanoliposomes technology. CAP-loaded nanoliposomes produced significant enhancement for anticancer activity against human cancer cell lines and selectivity compared to CAP and induced apoptosis in cancer cell lines in a dose-dependent manner. CAP-loaded nanoliposomes may serve as a promising approach for managing cancer and inflammation illnesses.

## Figures and Tables

**Figure 1 nutrients-13-03995-f001:**
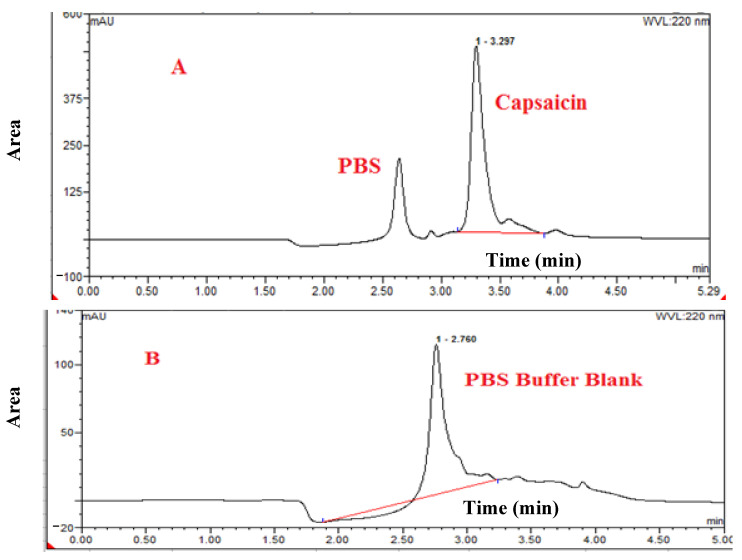
HPLC spectrum of (**A**): CAP loaded inside nanoliposomes, (**B**): PBS buffer blank hydration media.

**Figure 2 nutrients-13-03995-f002:**
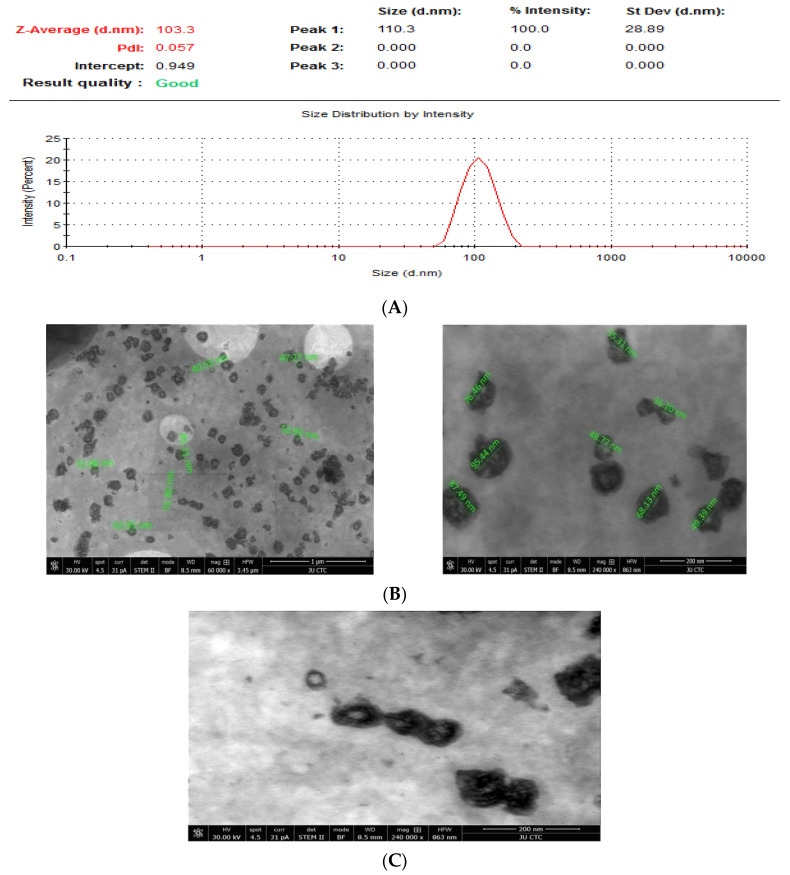
(**A**) Size distribution of CAP-loaded nanoliposomes by DLS; (**B**) TEM Shape and Size (CAP-loaded nanoliposomes); (**C**) Morphology (CAP-loaded nanoliposomes).

**Figure 3 nutrients-13-03995-f003:**
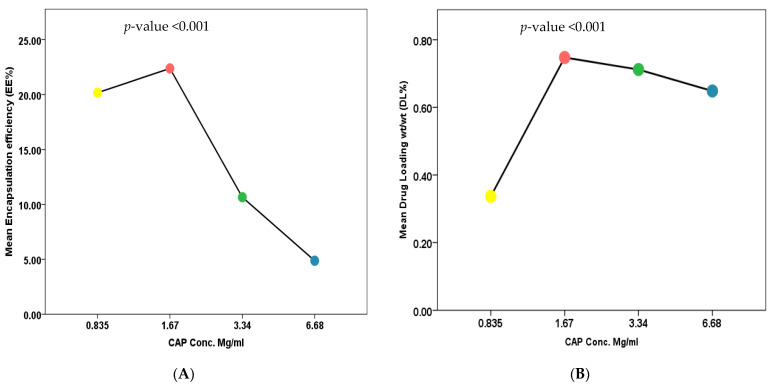
Effect of CAP amount on CAP-loaded nanoliposomes (**A**) %EE and (**B**) %DL.

**Figure 4 nutrients-13-03995-f004:**
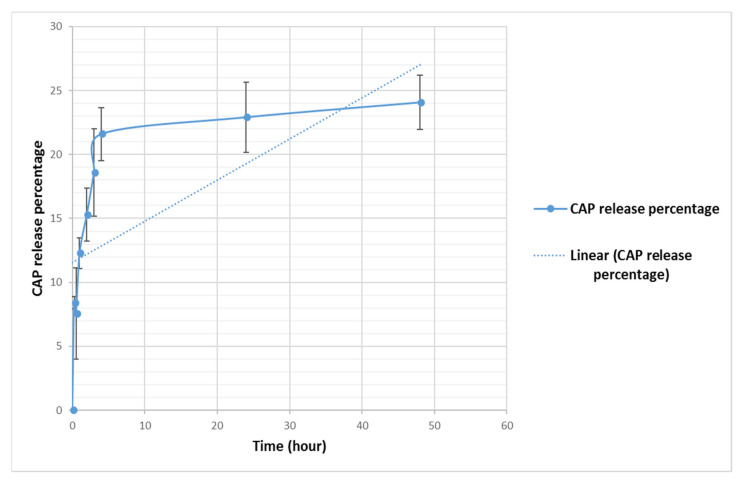
In vitro release of CAP from CAP-loaded nanoliposomes, monitored over 48 h at 37 °C. All values represent the average ± SD of three independent experiments.

**Figure 5 nutrients-13-03995-f005:**
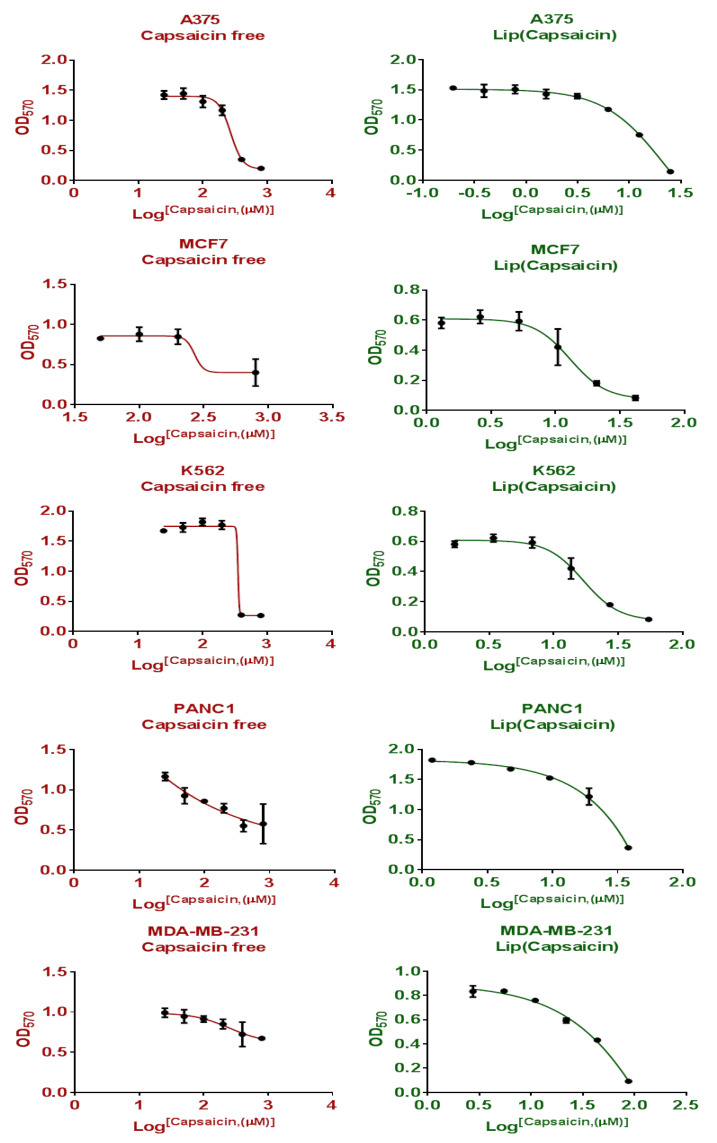
Cytotoxicity and the half maximal inhibitory concentration (IC_50_) of CAP and CAP-loaded nanoliposomes against MCF7 human breast cancer cell line.

**Figure 6 nutrients-13-03995-f006:**
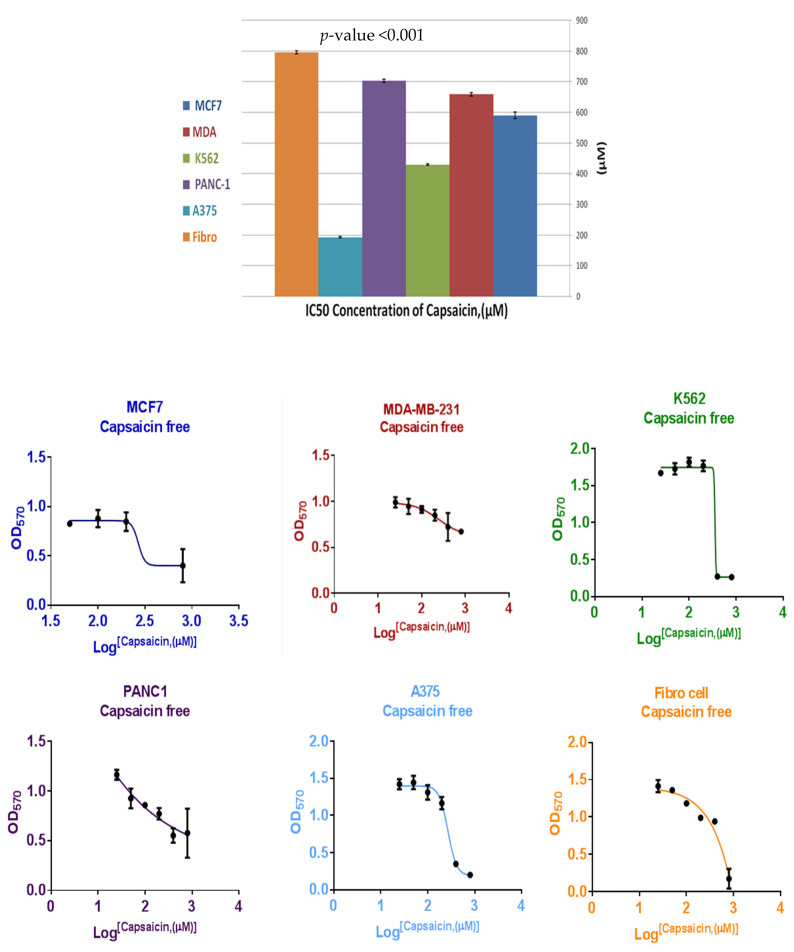
Cytotoxicity and the half maximal inhibitory concentration (IC_50_) of CAP against human fibroblasts normal cells as negative control and against human cancer cell lines (MCF7, MDA-MB-231, A375, K562 and PANC-1).

**Figure 7 nutrients-13-03995-f007:**
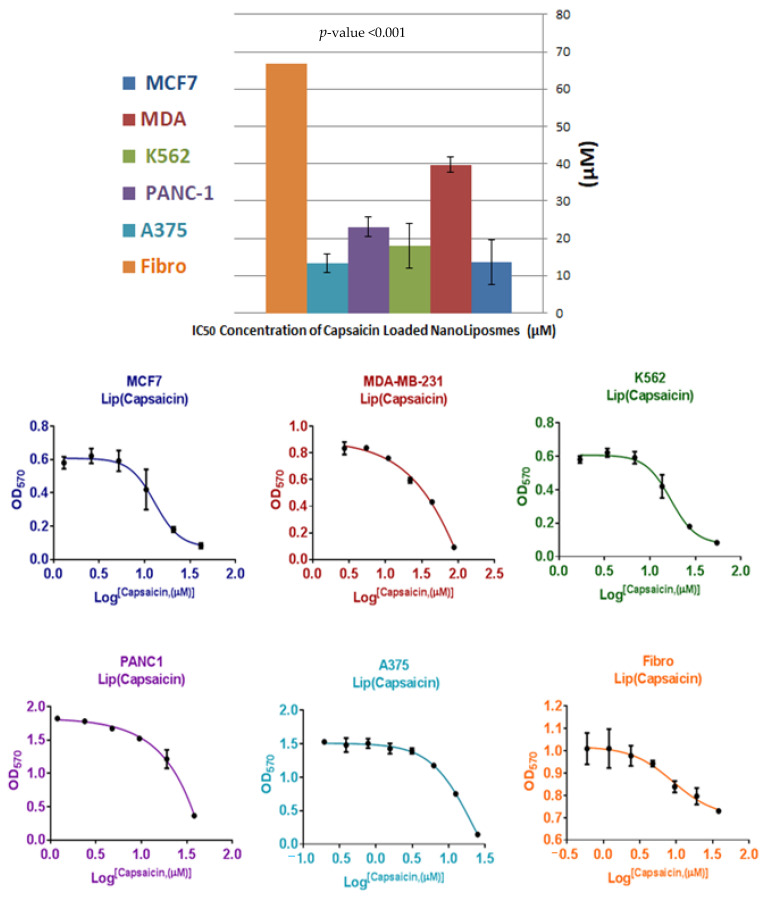
Cytotoxicity and the half maximal inhibitory concentration (IC_50_) of CAP-loaded nanoliposomes against human fibroblasts normal cells as negative control and human cancer cell lines (MCF7, MDA-MB-231, A375, K562 and PANC-1).

**Table 1 nutrients-13-03995-t001:** Effect of lipid ratio on CAP-loaded nanoliposomes (particle size, PDI, Zeta potential, and %EE).

Formula		Mean	Std.	Minimum	Maximum	ANOVA		Multiple Comparisons LSD		
						F	*p* Value	(I) F	(J) F	Sig.
Size	F # 1	99.48 *	3.90	94.57	106.40	6.23	0.006	F # 1	F # 2	0.009
	F # 2	118.54 *	14.18	98.91	130.50				F # 3	0.003
	F # 3	123.68 *	22.72	95.12	149.00			F # 2	F # 1	0.009
	-	-	-	-	-				F # 3	0.458
PDI	F # 1	0.11	0.04	0.05	0.15	0.811	0.455 ^NS^	-	-	-
	F # 2	0.13	0.04	0.09	0.19			-	-	-
	F # 3	0.12	0.02	0.09	0.16			-	-	-
Charge	F # 1	−12.44	0.87	−14.20	−11.00	3.04	0.064 ^NS^	-	-	-
	F # 2	−13.87	2.33	−17.00	−11.00			-	-	-
	F # 3	−14.61	1.99	−17.50	−12.20			-	-	-
%EE	F # 1	17.34	3.67	13.54	21.90	1.22	0.312 ^NS^	-	-	-
	F # 2	22.38	6.02	17.11	32.10			-	-	-
	F # 3	19.28	1.67	17.52	21.35			-	-	-

All data are normally distributed according to Shapiro–Wilk normality test. N ≥ 3; * Statistically different at a significance level, *p* < 0.05. ^NS^ Not statistically different at a significance level, *p* < 0.05.

**Table 2 nutrients-13-03995-t002:** Effect of CAP amount on (particle size, PDI, and Zeta potential).

ANOVA		Maximum	Minimum	SD	Mean	Formula	
*p*-Value	F						
0.328	1.2	130.5	98.91	14.2	118.54	F # 2	Size ^NS^
		192.3	127.8	30.91	155.68	F # 4	
		118.9	106.4	6.42	112.48	F # 5	
		132.3	95.97	19.02	114.18	F # 6	
		110.2	102.2	2.89	106.56	F # 7	
0.522	0.769	0.19	0.09	0.039	0.13	F # 2	PDI ^NS^
		0.58	0.09	0.19	0.251	F # 4	
		0.15	0.08	0.035	0.115	F # 5	
		0.15	0.1	0.017	0.121	F # 6	
		0.14	0.08	0.025	0.106	F # 7	
0.488	0.834	−11	−17	2.4	−13.87	F # 2	Charge ^NS^
		−10.2	−13.6	1.31	−12.01	F # 4	
		−13.6	−15.9	0.9	−14.56	F # 5	
		−11.3	−15.7	1.98	−12.91	F # 6	
		−12.6	−14.7	0.78	−13.68	F # 7	

All data are normally distributed according to Shapiro–Wilk normality test. N ≥ 3. ^NS^ Not statistically different at a significance level, *p* < 0.05.

**Table 3 nutrients-13-03995-t003:** Comparison of cytotoxic activities of CAP and CAPloaded nanoliposomes against human cancer cells and human normal fibroblasts cells.

		Mean (μM)	Std. Deviation	ANOVA		Multiple Comparisons LSD		
				F	*p*-Value	(I) F	(J) F	*p*-Value
CAP-loaded nanoliposomes	MCF7	13.66 *	5.97	76.83	<0.001	Fibro	MCF7	<0.001
	MDA	39.72 *	2.02				MDA	<0.001
	K562	17.88 *	5.97				K562	<0.001
	PANC-1	23.04 *	2.60				PANC-1	<0.001
	A375	13.32 *	2.53				A375	<0.001
	Fibro	66.77 *	3.70				-	-
CAP	MCF7	590.81 *	10.15	4488.32	<0.001	Fibro	MCF7	<0.001
	MDA	659.35 *	6.03				MDA	<0.001
	K562	430.21 *	5.09				K562	<0.001
	PANC-1	703.87 *	3.16				PANC-1	<0.001
	A375	193.50 *	2.53				A375	<0.001
	Fibro	796.28 *	3.03				-	-

* Statistically different at a significance level, *p* < 0.05.

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
