# Peer review of "Preparation, Characterization, and Anticancer Effects of Capsaicin-Loaded Nanoliposomes"

_nutrients, 2021, doi:10.3390/nu13113995_

Round 1
Reviewer 1 Report
Please consider the following comments:
- Line 125 states that "seven different formulae were prepared as shown in Table 1," however, the Table 1 only presents 5 formulae. This needs to be explained.
- The statment in lines 129-131 is not very clear. Explain the composition of DPPC:Cholesterol:DSPE:PEG2000 in F4-F6 nanoliposomes.
- The procedure for drug release as described in lines 153-157 is not clear. This needs to be rewritten for clarity.
- Table 3: The data for CAP-loaded liposomes and CAP needs to be separated by a line to make distinction between cells treatment in each group.
Author Response
We would like to thank for the reviewer 1 for commenting on our manuscript and the time spent to read our work. attached file explains point by point our response to the concerns raised.

Reviewer 2 Report
The aim of the paper is to investigate the effect of capsaicin loaded into nano-liposomes and testing its anticancer activity against MCF7, MDA-MB-231, K562, PANC1, and A375 cell lines. The manuscript is organized and well written, and has the potential for novelty, as this is the first report on the anti-cancer activity of such a formulation against several cancer cell types in vitro. The conclusions are consistent with the results obtained and with the data already existing in literature. The findings are also well discussed. However, there are still many small but important things to be improved in the manuscript. In my opinion, the submission requires major edition and improvement at some points, main of which are listed below.
L17: “capsaicin” written with a lowercase letter. Please check the whole manuscript thoroughly.
L3, 20: Authors should decide whether to write the word "nanoliposomes" intermittently (nano-liposomes) or continuously (nanoliposomes) and use one spelling of the word throughout the manuscript.
L70: Latin names should be italicized, e.g. Capsicum annuum.
L125: should be: in Table 1 and 2.
L139, 141, etc.: Authors should pay attention to the interval between the numerical value and the unit, e.g. 200 μL, 0.45 μm, etc.
L165: Please increase the interval between the next subsection "d".
L192, 241, 245, etc.: Authors should use the symbol "p" for the significance level in lower case and additionally in italics, e.g. p < 0.05 or p = 0.278 throughout the manuscript.
L214: should be α = 0.05. Please check the whole manuscript thoroughly.
L218: If a new abbreviation is introduced in the text, please explain it, despite the list of abbreviations at the end of the manuscript. The same applies to the cancer cell line abbreviations used in the research - please explain them in the text where they were first used.
L214-216: Please always use units for numerical values, e.g. (99.48 ± 3.90 nm).
L224, 234: In Tables 1 and 2, in the column (Mean) with the values for individual parameters (separately for size, PDI, Charge,% EE), please enter statistical markings that allows to determine whether the results are statistically different at a significance level, e.g. p < 0.05.
L290-292: should be “IC50”. Please check the whole manuscript thoroughly.
L302-304: This sentence is not grammatically correct.
Figure 7: The title of figure 7 should concern the Lip(capsaicin). Similarly, the caption under the bar chart.
L425: Please remove the lines in the “List of abbreviations” table.
Author Response
We would like to thank this reviewer for his appreciation of our work and taking the time to correct our manuscript. Attached is a point by point our response to the concerns raised.
